# NEURAL COHERENCE

## ABSTRACT

Many important machine learning problem settings involve adapting a large pre-trained model to a limited number of examples. To create a good pre-trained model a number of choices must be made, from determining when to stop training the model, to which dataset(s) to train the model upon. In this work we develop a principle that we shall refer to as Neural Coherence, which is based on characterizing the way in which a neural network behaves on different types of inputs using the statistics of activation functions across different sets of inputs. Our experiments show that these measures of Neural Coherence can be used to formulate a general approach for inferring and improving the generalization of a model to a downstream problem. We show how Neural Coherence can be used to make decisions for early-stopping, and to infer which dataset to train on, given a target dataset. Overall, our experiments indicate that our approach to using Neural Coherence for model selection tasks can significantly improve the performance of deep learning models to out-of-distribution downstream problems.

## 1 INTRODUCTION

In recent years, the practice of using models pre-trained on datasets that are known to generalize well to other tasks has become the de facto standard for attaining high performance on various downstream tasks [Tan & Le, 2021; Liu et al., 2022; Woo et al., 2023]. This paradigm offers a compelling advantage: the ability to utilize large, powerful models when downstream target data is scarce, which is a true constraint for many real-world problems [Raghu et al., 2019]. Furthermore, on a larger scale, this also allows significant computational resources to be expended once to create a foundation model that is subsequently used by many others for other tasks. Consequently, a key challenge lies in effectively transferring and adapting the representations of pre-trained models to the specific downstream problem at hand, so that downstream performance is maximized.

Since downstream tasks are frequently out of distribution (OOD) regarding the original training data, maximum downstream performance (Source-Val) may not coincide with the maximum validation set performance of the original dataset for the pre-trained model [Gulrajani & Lopez-Paz, 2021], making decisions like selecting the optimal pre-training data or checkpoint selection during pre-training difficult. This situation is further complicated when target annotations are not available for the task at the time or are expensive and slow to obtain in sufficient quantities, bottle-necking model development.

In this paper, we propose a novel approach for guiding such decisions. Our method is based on the coherence of activation statistics and we refer to these types of statistical quantities and our approach as Neural Coherence (NC). Figure 1 illustrates our Neural Coherence principle. In our particular implementation of the principle, we characterize neural coherence using statistical moments of neural activations on training data and small amounts of unlabeled target task data. Through our experiments, we demonstrate that neural coherence can be used effectively on large models to guide important choices during the development process. As one example, **we show that neural coherence performs well even when data is limited to less than 20 unlabeled examples** on checkpoint selection during pretraining. We also show that pretraining datasets can be selected with neural coherence similarly, highlighting its versatility.

We exemplify the usefulness of this method on two example tasks: checkpoint selection during pre-training and pre-training dataset selection. We show that our method works consistently and reliably, outperforming other baseline methods, especially when data is scarce, unlabeled and the model has not been trained for a fraction of the entire training steps.

We validate our finding through numerous ablations studies on a wide variety of complex datasets starting with ImageNet1K [Russakovsky et al., 2015], and fine tuning to: Food101 [Bossard et al., 2014], and EuroSat [Helber et al., 2017].

In short, our contributions are as follow:

- We introduce Neural Coherence (NC), a method for analyzing the generalization of features on target datasets with scarce data.
- We show that NC can be applied on large, state-of-the-art models such as ConvNeXt L.
- We demonstrate the usefulness of neural coherence to guide critical decision during pre-training, such as checkpoint selection and training dataset selection.
- We show that our method works particular well in scenarios when (labeled) data is scarce for the downstream tasks.

## 2 OUR NEURAL COHERENCE PRINCIPLE

Assume a target problem $T$ where performance has to be maximized, i.e. we want a model $f$ with minimal target empirical risk $\mathcal{L}_T$. Given a set of candidate models $f$ trained on a source problem $S$, obtained by varying a hyper-parameter $\Omega$ and ordered by decreasing source empirical risk $\mathcal{L}_S$, Neural Coherence characterizes the trajectories $\psi_T$ and $\psi_S$ of the neural activation distributions $p(\mathbf{z})$ for the source and target data $\mathbf{x}_S$ and $\mathbf{x}_T$. Notably, Neural Coherence gives a measure of how the two trajectories are evolving in a similar direction, with stronger Neural Coherence equating to more similar directions. We then hypothesize that, starting from the beginning of the model sequence ordered by decreasing $\mathcal{L}_S$, the target empirical risk $\mathcal{L}_S$ will also decrease, as long as the target and source trajectories remain coherence. Target performance would either reach its optimum before the trajectories start to diverge, or at the end of the model sequence if no divergence occurs.

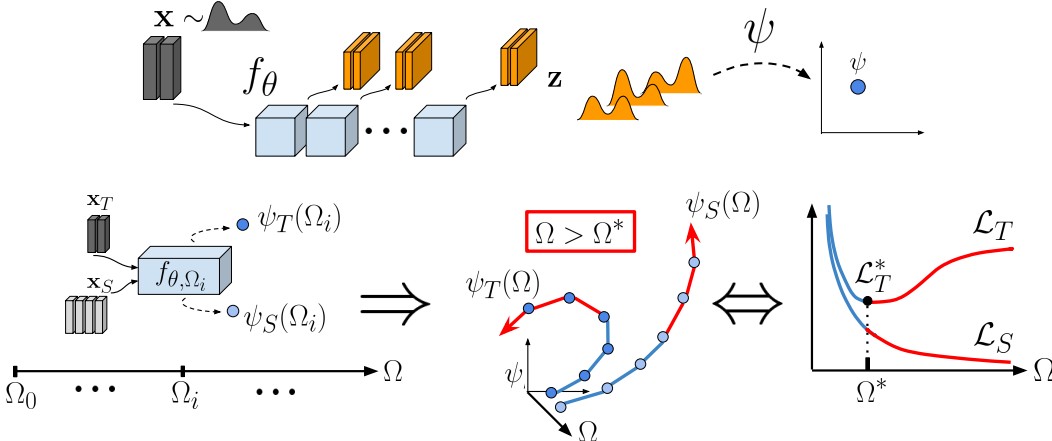

Figure 1: Given a target problem, and a set of models obtained by varying a hyper-parameter $\Omega$ and ordered by increasing source performance, Neural Coherence compares the trajectories of the neural activation distributions for the source ($\psi_S$) and target ($\psi_T$) data, and predicts that the model at the end of the coherent phase of the trajectories should yield the best target performance. This transition occurs when the trajectories turn from blue to red. We test the hypothesis that the source and target losses ($\mathcal{L}_S$, and $\mathcal{L}_S$ respectively, shown at the bottom right) will also begin to diverge at this point.

The following five concepts are central to our formulation of Neural Coherence as a quantity that one can compute, and as an approach to model selection:

*Full network activations:* We analyze activations across the entire network (rather than simply final representations or outputs).

*Activation distributions embedded into a vector space:* We embed the activation distribu-

tions into a vector space that, while being simple, is rich enough to capture most important dynamics (rather than a narrowly defined statistical metric).

*Activation trajectories:* We characterize the trajectory that activation distributions undergo along the considered hyper-parameter. This allows us to leverage more information about learning dynamics, compared to analyzing a single state.

*Source-Target comparisons:* We compare the trajectory of the target activations with that of the source activations (instead of either just looking at the source or the target).

*Trajectory coherence:* Once we having trajectories, we estimate when their coherence starts to collapse and start to diverge. Our central hypothesis is that this should reflect the target vs. source performance dynamics and we examine that hypothesis in our experimental work below.

## 2.1 PRACTICAL IMPLEMENTATION

To implement Neural Coherence, we first need to characterize the distribution of neural activations $\mathbf{z} = \varphi_l(\mathbf{x}; t)$. Because many distributions have shown to be fully determined by their sequence of moments [Yarovaya et al., 2020], we use the moment sequence to characterize the neural activation distributions. Recall that, for an arbitrary random variable $Z$ with a probability distribution $p(z)$, its n-th order moment is defined as $\mathbf{m}_n(z) = \mathbb{E}_{p(z)}[z^n]$. For a multivariate random vector $\mathbf{z}$, its sequence of moments starts with the mean vector $\mathbf{m}_1(\mathbf{z}) = \mathbb{E}_{p(\mathbf{z})}[\mathbf{z}]$, followed by the autocorrelation matrix $\mathbf{m}_2(\mathbf{z}) = \mathbb{E}_{p(\mathbf{z})}[\mathbf{z}\mathbf{z}^\mathrm{T}]$, and so on.

For tractability, and because only few target samples are available, we limit sequence estimation to the second order moment, avoiding the excessive standard error of higher order statistics. To reduce the signal-to-noise ratio, we further aggregate those statistics by computing their moments across the $D$ feature dimensions. This dimensionality reduction also renders our method computationally feasible. To avoid resulting higher order statistics, we only computes the 1st-order feature-wise moment over $\mathbf{m}_2(\mathbf{z})$. To preserve meaningful information, the diagonal and non-diagonal elements of $\mathbf{m}_2$ are aggregated independently. We obtain $\hat{m}_1$, $\hat{m}_2$, $\hat{m}_3$ and $\hat{m}_4$, the *aggregated moments*, defined in Eq.1:

$$\hat{m}_1(\mathbf{z}) = \frac{1}{D}\sum_i^D \mathbf{m}_1^i(\mathbf{z}) \qquad \hat{m}_2(\mathbf{z}) = \frac{1}{D}\sum_i^D \left(\mathbf{m}_1^i(\mathbf{z})\right)^2$$

$$\hat{m}_3(\mathbf{z}) = \frac{1}{D}\sum_{i=j}^D \mathbf{m}_2^{i,j}(\mathbf{z}) \qquad \hat{m}_4(\mathbf{z}) = \frac{1}{D}\sum_{i\neq j}^D \mathbf{m}_2^{i,j}(\mathbf{z}) \tag{1}$$

where $\mathbf{m}_1^i(\mathbf{z})$ is the i-th feature of $\mathbf{m}_1(\mathbf{z})$, and $\mathbf{m}_2^{i,j}(\mathbf{z})$ is the feature at the i-th row and j-th column of $\mathbf{m}_2(\mathbf{z})$. We then characterize the trajectory of neural activation distributions as the tensor $\psi(\mathbf{x}; \Omega)$, defined in Eq. 2 :

$$\psi(\mathbf{x}; \Omega)_{l,k} = \hat{m}_k(\varphi_l(\mathbf{x}; \Omega)) \tag{2}$$

where $l \in [1, L]$ and $k \in [1, 4]$. Then, for each layer $l$ and aggregated moment $k$, we find the hyper-parameter value $\hat{\Omega}_{l,k}$ where the trajectories $\psi(\mathbf{x}_S; \Omega)_{k,l}$ and $\psi(\mathbf{x}_T; \Omega)_{k,l}$ start to diverge from each other (which will be the last value of $\Omega$ if no divergence occurs at all). To find $\hat{\Omega}_{l,k}$, we first, compute the linear regression of each trajectory, which gives us the average direction it is headed to. We then compute the product between the two slopes, where a negative product indicates that trajectories are going in opposite directions. To quantify coherence, we consider the main direction each trajectory is undertaking, by computing the linear approximations of the trajectories. We obtain the linear regression parameters $a$ and $b$ (slope and intercept) by minimizing the mean squared error (MSE) :

$$a, b = \operatorname*{argmin}_{a,b} \frac{1}{(\Omega_2 - \Omega_1)} \sum_{\Omega=\Omega_1}^{\Omega_2} \left((a\Omega + b) - \psi(\mathbf{x}; \Omega)_{l,k}\right)^2 \tag{3}$$

We obtain $a$ and $b$ analytically by solving the ordinary least square. Given a pair of source and activation trajectories $\psi(\mathbf{x}_S; \Omega)_{l,k}$ and $\psi(\mathbf{x}_T; \Omega)_{l,k}$, within an interval $[\Omega_1, \Omega_2]$, we compute the linear regression parameters $a_S, b_S$ and $a_T, b_T$. We define our objective function $d_{NC}$ for the

divergence as the angle between the two regression lines, multiplied by the size of the interval:

$$\text{NC}(\mathbf{x}_B, \mathbf{x}_T; l, k, \Omega) = \cos^{-1}\left(\frac{[1, a_S] \cdot [1, a_T]}{\|[1, a_S]\|\|[1, a_T]\|}\right) \times (\Omega_2 - \Omega_1) \tag{4}$$

## 2.2 Application to checkpoint selection and training data selection

**Checkpoint selection**   When analyzing a neural network, the latent representations of each layer create an individual trajectory for the target OOD-data and the in-distribution source data. For architectures with four or fewer layers Guiroy et al. [2022] has shown that simply focusing on the layer with the strongest divergence can be sufficient for determining early stopping time.

While focusing on the strongest divergence has been demonstrated to work in small scale neural networks with few hidden layers, we find it insufficient when working with very deep models with hundreds of layers, such as ConvNeXt. We find early-stopping decisions unreliable, when a single layer indicates a strong divergence while hundreds more show no divergence between the target and source activations. Moreover, in very deep models, multiple layers might show significant divergence. Thus the question arises of how to weight their contributions to make the early-stopping decision $t^* = \sum_l \alpha_l \operatorname{argmax}_t d_{t,l} = \sum_l \alpha_l t_l^*$. We propose a weighting function of the contributions of each layer, based on their individual divergence scores, where layers showing strong divergence have more influence over the final early-stopping decision $t^*$, while layers showing little divergence have little influence. The score vector must be normalized such that such that it sums to one.

We consider the following normalized scores : $\alpha_l = \frac{\text{NC}_l}{\sum_l \text{NC}_l}$ and our resulting stopping epoch $\hat{t}_{\text{NC}}$ is defined as :

$$\hat{t}_{\text{NC}} = \sum_l \alpha_l \times \operatorname*{argmax}_{t_1} \text{NC}(l, k; \ t_1 \le t \le t_{valid}^*) \tag{5}$$

**Training data selection**   We can set the hyper-parameter to be optimized as data distribution $p(\mathcal{D})$ used to train the model. In a simple case, given two candidate distributions $p_A$ and $p_B$, we can define the line interpolating between $p_A$ and $p_B$ as $p(\mathcal{D}; \Omega_{AB}) = \Omega_{AB} p_A + (1 - \Omega_{AB}) p_B$ such that $\mathbf{x} \sim \Omega_{AB} p_A(\mathbf{x}) + (1 - \Omega_{AB}) p_B(\mathbf{x})$ where $\Omega_{AB} \in [0, 1]$.

Here the Neural Coherence is thus measured along this line, and used to estimate the optimal value for $\Omega_{AB}$. In this work, given two candidate distributions $p_A$ and $p_B$, we focus our experiments in the simpler case of finding the best candidate among those two. However, given a set of candidate distributions, of size larger than two, we apply this approach sequentially to perform pair-wise comparisons and estimating the best candidate distribution among the set, which we apply in our experiments.

However, dealing with training data selection poses some extra difficulty. As opposed to an hyperparameter like training time, here the order over the set of models is not given implicitly. In other words, there is no trivial way to know if going from $p_A$ to $p_B$ necessarily improves target generalization up to a point, or if it should be the other way around. This problem is exacerbated when the optimum lies at either extremity of $\Omega_{AB}$.

For a more robust interpretation of whether the trajectories are coherent or divergent, along a given direction, we propose to compare their coherence between the forward and backward direction. In other words, along the forward direction between $p_A$ and $p_B$, we measure the neural coherence between the activations $\psi_T$ and $\psi_B$, and along the backward direction, we measure the coherence between $\psi_T$ and $\psi_A$. Hence, if the forward direction shows a stronger neural coherence than the backward direction, we infer that training on dataset B yields a better generalization to the target problem, and vice versa.

Selected training distribution from Neural Coherence :

$$p^*(\mathcal{D}) = \{p_B \ \text{ if } \ \text{NC}(\mathbf{x}_B, \mathbf{x}_T; \Omega_{A,B}) > \text{NC}(\mathbf{x}_A, \mathbf{x}_T; \Omega_{B,A}), \ \text{ otherwise } \ p_A\} \tag{6}$$

## 3 EXPERIMENTS

We move on to evaluate the performance Neural Coherence. First, we discuss the methodology and experimental setup, followed by experiments on checkpoint selection, training data selection, and ablation studies on the statistical efficiency of Neural Coherence.

### 3.1 CHECKPOINT SELECTION

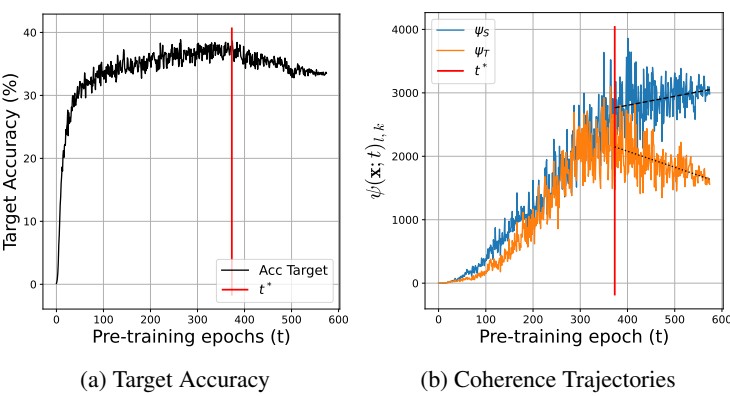

(a) Target Accuracy      (b) Coherence Trajectories

Figure 2: Qualitative demonstration of Neural Coherence for checkpoint selection. A ConvNeXt-L is trained on ImageNet1k, with ImageNet-Sketch [Wang et al., 2019] as target. The target trajectory is computed from 5 samples while the source trajectory uses the whole source validation split.

In our experiments we focus on ConvNeXt-L [Liu et al., 2022], a modern state-of-the-art architecture, which has around 200 million parameters with more than two hundred hidden layers. The ConvNeXt family furthermore incorporates aspects of ResNet and Transformer architectures, making it a good indicator for the applicability of NC . We train ConvNeXt-L using the setup of the original work [Liu et al., 2022].

To demonstrate the performance of NC , we consider two different OOD paradigms : zero-shot generalization and transfer-learning. For each paradigm, we evaluate the performance of NC on two different target datasets. In each paradigm, target datasets fall into two scenarios: In Scenario A, the best performance on the target dataset is achieved before the final epoch of training. In other words, neural coherence breaks down during training, and we are testing how well our method can detect this breakdown accurately. Scenario B represents the setup where final epochs also provide the best performance - i.e. neural coherence never breaks down. In this case, we expect NC to pick the latest checkpoint during training. With Scenario B, we want to demonstrate that NC is not biased towards picking earlier checkpoints. This is important as a breakdown of neural coherence is not guaranteed for real-world scenarios.

As baseline methods for comparison, we use: 1) In-Distribution Validation (Source-Val), where we simply select the checkpoint with the highest accuracy on the entire validation set; 2) Few-Shot-Generalization-Estimation (Target-Val), where this baseline uses $n$ fixed labelled examples to approximate the target generalization such that $n$ matches the fixed number of unlabelled examples from the target problem that NC uses; 3) ABE [Guiroy et al., 2022], an early stopping technique also working activation statistics of a number $n$ of unlabeled examples 4) NCE [Tran et al., 2019] 5) PARC [Bolya et al., 2021] as additional model selection baselines.. In each experiment, to give context on the obtained performance, we show the maximum performance that was achievable if an *Oracle* could stop at the time of maximum target accuracy.

**Zero-shot generalization**    We investigate the capabilities of NC in a simple zero-shot generalization setting using two pairs of datasets. We consider Imagenet-A [Hendrycks et al., 2021b] and Imagenet-Sketch [Wang et al., 2019] as target datasets for our ImageNet1k-pretrained model. This allows for a simple zero-shot-generalization setup, since all datasets have the same classes as ImageNet1k (except ObjectNet, for which we use the set of 113 overlapping classes with ImageNet1k). In this paradigm,

the model performs no fine-tuning, and its performance is directly evaluated as its average accuracy on the test split of the target dataset (zero shot generalization). For NC , ABE and Target-Val, we run 250 independent trials, where each trial uses a fixed set of $n = 5$ target examples. The Source-Val baseline has access to the entire validation split of Imagenet. For each experiment, the performance is computed on the full test split of the target dataset, and averaged over all the trials. The Oracle correspond to the maximum possible target accuracy, i.e. if an oracle knew exactly when to stop.

Table 1: Zero-Shot Generalization - Performance of NC (Top-1 Accuracy)

| | **Scenario A:** max $Acc_{target}$ happens before max $Acc_{valid}$ | **Scenario B:** max $Acc_{target}$ does *not* happen before max $Acc_{valid}$ |
|---|---|---|
| | ImageNet-S | ImageNet-A |
| *Oracle* | 37.72 | 14.58 |
| Source-Val | 35.75 | 14.58 |
| Target-Val | 30.96 | 9.55 |
| PARC | 23.76 ± 3.12 | 9.74 ± 0.85 |
| NCE | 0.92 ± 0.00 | 3.91 ± 1.12 |
| ABE | 24.66 ± 0.83 | 8.34 ± 0.68 |
| **Neural Coherence** | **36.34** ± 0.04 | 14.45 ± 0.01 |

From Tab. 1, we observe that NC is significantly outperforming Source-Val, ABE and Target-Val, being significantly closer to the oracle-performance. These results show that our method is able to locate the breakdown of neural coherence. We also demonstrate that these breakdowns are indicative of for feature generalization. In Scenario B, we observe that NC performs close to optimal, demonstrating the robustness of our method in scenarios where Source-Val is optimal and neural coherence never breaks down as a result. In fact, our method is more robust than Target-Val and ABE, which performed significantly worse under those circumstances.

**Linear Probing**   Linear probing is one of the most popular methods transfer-learning, only training a linear classifier while freezing the feature extractor. Since this method also only requires the training of a single, linear layer. Linear probing is also highly data efficient, which is important for scenarios with scarce data - the intended use-case of NC. Additionally, since we freeze the feature extractor and the trained part of the model is linear, the model optimization is a convex problem. In combination, this allows us to obtain a clean signal on the quality of selected feature extractors [Alain & Bengio, 2017] and by extension, a good measure of quality for our selection methods. Note that because the classes are different between the target and source domains, this does not allow to directly compare NC with the Target-Val baseline. Please refer to the Appendix for this comparison.

We use Imagenet1K as the source dataset. Our model achieved a state-of-the-art performance of 84.4%, close to the result of the original authors [Liu et al., 2022]. We use FOOD-101 [Bossard et al., 2014], and EuroSat [Helber et al., 2017] as target datasets.

Table 2: Linear Probing - Performance of NC (Top-1 Accuracy)

| | **Scenario A:** max $Acc_{target}$ happens before max $Acc_{valid}$ | **Scenario B:** max $Acc_{target}$ does *not* happen before max $Acc_{valid}$ |
|---|---|---|
| | FOOD-101 | EuroSat |
| *Oracle* | 44.45 | 74.87 |
| Source-Val | 41.72 | 74.87 |
| PARC | 35.16 ± 0.36 | 34.43 ± 0.53 |
| ABE | 40.17 ± 0.62 | 63.47 ± 0.11 |
| **Neural Coherence** | **43.80** ± 0.06 | 74.53 ± 0.02 |

In Tab. 2, similarly to the zero-shot experiments, **NC outperforms ABE and Source-Val in Scenario A, even though Source-Val has access to the full validation set while NC is only using 5 unlabeled examples**. For FOOD-101 and EuroSat, NC respectively close 76% and 82% of the generalization

gap that Source-Val suffers. In Scenario B (EuroSAT), NC outperforms ABE and is almost on par with Source-Val. Also demonstrating the robustness of NC in a scenario where a part of the model is trained on novel data. In summary, the results demonstrate that NC selects high-quality pre-training checkpoints reliably in all tested scenarios.

## 3.2 Pre-Training data selection

The dataset used for pretraining ultimately determines the features the model will learn. Hence, the choice of pretraining data is a relevant task in scenarios where we have to rely on pretrained features. Source performance on the pre-training set isn't guaranteed to correlate with the target generalization. Thus, a more elaborate model selection scheme like NC is necessary. In this section, we will demonstrate that NC can be used for selecting pre-training data. We achieve this, by applying our method on models pre-trained for 10% of a full pretraining steps on various datasets and combinations of datasets. For our experiments, we use few-shot image classification datasets, with 5-way 1-shot tasks. We use the MAML algorithm for training the models. We use the standard 4-layer ConvNet architecture proposed by [Vinyals et al., 2016]. We rely on a set of five diverse image datasets, for our target problems : Mini-Imagenet, Omniglot, Quickdraw, Traffic Sign, Describable Textures. For pre-training, the agent can select any combination of those five datasets. Therefore, the challenge lies in selecting, the model pre-trained on a dataset or combination of datasets that is optimal for the downstream task. We demonstrate in Tab. 3 the performance of the Neural Coherence approach in predicting the best model depending on the pre-training dataset, outperforming source validation in all scenarios. While the reliability of NC has a non-negotiable variance between datasets, we significantly beat the chance level of 25% predictor in all cases. Furthermore, it is worth keeping in mind that these decisions were made based on severely undertrained models (10% of the full training). This is important, as a full training on any combination of potential pretraining datasets would be impractical.

| | **Target Dataset** | | | | |
|---|---|---|---|---|---|
| **Method** | Mini-Imagenet | Omniglot | Quickdraw | Traffic Sign | DTD |
| *Achieved target performance* | | | | | |
| *Oracle* | 46.1 | 86.2 | 55.6 | 90.6 | 32.8 |
| Source-Val | 29.4 | 72.8 | 55.6 | 37.7 | 26.0 |
| **Neural Coherence** | **33.5 ± 0.1** | **82.7 ± 0.9** | **55.6 ± 0.1** | **41.4 ± 0.6** | **32.7 ± 0.1** |
| *Accuracy of Neural Coherence in predicting the best source dataset* | | | | | |
| Acc-1 (chance=25%) | 88.0 ± 5.2 | 74.0 ± 7.0 | 98.7 ± 1.8 | 42.0 ± 7.9 | 98.7 ± 1.8 |

Table 3: Training data selection. Model: Standard CNN; Optim: MAML; Datasets : Mini-Imagenet, Omniglot, Quickdraw, Traffic Sign, Describable Textures (DTD). For each target dataset, the task consists of predicting which of the remaining four constitutes the best dataset to use as the training distribution. Number of target examples per trial : 5. Each experiment is repeated at least over 50 trials, or until obtaining tight confidence intervals.

## 3.3 Ablation : Statistical efficiency

In this section, we show that NC consistently requires fewer examples than the baseline of Few-Shot Generalization Estimation (Target-Val), to achieve a similar level of downstream performance. We repeat the experiment described in section 3.1 comparing only NC and Target-Val. We evaluate the performance of both checkpoint selection methods for batch sizes $n = \{1, 2, 3, 4, 5, 20\}$ to assess the ablation of predictive performance of the selected checkpoint as a function of the set of randomly chosen samples $n$. On both datasets used for this experiment we fined that NC is outperforming Target-Val (See Fig. 3). **In fact, in all tested scenarios, NC using only a single unlabeled sample is outperforming Target-Val** using up to 20 labeled samples, highlighting the statistical efficiency of NC .

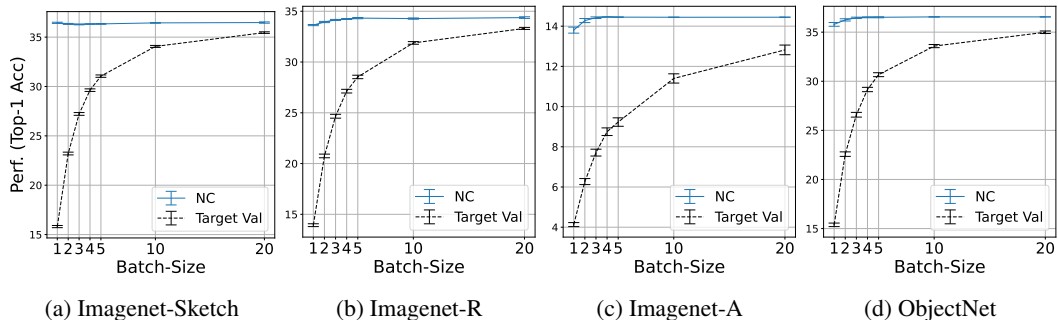

|                | (a) Imagenet-Sketch | (b) Imagenet-R | (c) Imagenet-A | (d) ObjectNet |

Figure 3: Statistical efficiency of NC in low data regimes : We see that, for a same number of examples, NC consistently outperforms Few-Shot Generalization Estimation (Target-Val) in estimating the appropriate checkpoint.

## 4 RELATED WORK

The problem of model selection for out-of-distribution domains and problems has been well moti-vated, including applications such as checkpoint selection / early-stopping [Goodfellow et al., 2016; Gulrajani & Lopez-Paz, 2021], [Fisch et al., 2023]. Many subfields of machine learning, such as meta-learning, typically aim at developing optimizers or models capable of easy and efficient adaptation to new tasks [Finn et al., 2017; Snell et al., 2017; Vinyals et al., 2016]. In general, there are three options when estimating the performance of a given pre-training setup on a target dataset. First, we can assume that an in-distribution performance validation (Source-Vall) is well correlated with the target domain, which is not guaranteed. The second option is to validate the performance on the target domain (Target-Val), which is dependent on the presence of sufficient data and labels. A third option is to utilize heuristics, which allow for better model selection in scenarios where the aforementioned methods would provide insufficient or suboptimal results.

A common strategy for assessing the efficacy of such models is to rely on an in-distribution-validation (Source-Val) set to stop the pre-training early or select a checkpoint for finetuning [Xie & Richmond, 2018; Hoffmann et al., 2022; Zhang et al., 2021; Bonet et al., 2021; Yao et al., 2022]. Those methods have the major limitation that they are not informed by knowledge about the target domain or problem, thus potentially suffering from a generalization gap. Indeed, the authors of [Gulrajani & Lopez-Paz, 2021] also argue that model selection methods for Out-of-Distribution and domain generalization must be informed, in some capacity, by the actual downstream domain, if they are to be effective.

Different works have proposed heuristics for estimating model generalization to out-of-distribution domains and performing model selection. They often revolve around the idea of measure some statistics on the model representation. Some are independent of the target domain, but such methods, just like source validation, can be suboptimal due to the potential distributional between the source and target domains. Other approaches consider the target domain [Tran et al., 2019; Guiroy et al., 2020]. We found that most heuristics of the sort tend to use some of the five core elements constitutive of Neural Coherence, such as analyzing model representations, inspecting activations across the entire network, or considering a broader family of statistical distances. However, to the best of our knowledge, none have used all those elements in conjunction, even less so the use of trajectories of model states nor the concept of trajectory coherence, as scalar distance functions are predominant [Bolya et al., 2021; Sun & Saenko, 2016; Dwivedi et al., 2020; Dwivedi & Roig, 2019; Kornblith et al., 2019; Raghu et al., 2017].

Other work have studied the relationship between OOD generalization and latent dynamics of neural networks. The authors of [Yosinski et al., 2014] show that the ability of neural networks for knowledge transfer can be related to important intermediate hidden-layers. Other works study OOD generalization in the context of Meta-Learning or Few-Shot Learning. The authors of Raghu et al. [2020] observed that when fine-tuned on a new task, while the linear classification head changes drastically while backbone remains approximately invariant. In Goldblum et al. [2020] the authors observed that generalization might be related to a notion of representation clustering. On the other

hand, Dhillon et al. [2020] and Frosst et al. [2019] observed in OOD settings, the clustering of final representations might not be indicative of target generalization.

## 5 CONCLUSION

In this work, we propose *Neural Coherence* (NC), an approach for selecting the appropriate checkpoint for OOD-generalization to a specific target dataset. NC requires as little as five unlabeled data points while still outperforming in-distribution-validation and few-shot-generalization-estimation baselines. Through experimental evaluation, we show that NC is working reliably on various target datasets as well and within zero-shot and transfer-learning setups.We also empirically demonstrate the statistical efficiency of NC against few shot generalization estimation (Target-Val).

Overall, this work further motivates the practice of careful model selection for OOD generalization, even for large foundation models. We believe this result is of considerable practical significance to machine learning researchers and practitioners as our proposed algorithm can potentially improve downstream performance of foundation models. Finally, this work further motivates the neural coherence principle and future research around it. Because of this, we believe that many interesting future directions are warranted, of which we make two concrete suggestions: First, investigating how to apply NC to other data domains. Second: the application of neural coherence principles to infer other important hyperparameters than training time.

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

## A  ADDITIONAL EXPERIMENTS

In this section, we introduce tables containing additional results that further verify our finding in the main paper. For checkpoint selection in the zero-shot setting we select to additional dataset. ImageNet-R [Hendrycks et al., 2021a] and ObjectNet [Barbu et al., 2019], since these share the same classes as the other datasets tested so far, making the zero-shot evaluation setup trivial. In Table 4, we see that also on these additional datasets, we are able to outperform ABE. In case Source-Val is optimal (Scenario B) we are close to the optimal solution, while in Scenario A, we are outperforming all competing baselines. For the linear probe scenario, we choose the iNaturalist [Horn et al., 2017] and PlantNet300K [Garcin et al., 2021] datssets, as these to also mark a significant domain shift from ImageNet1k. In these scenarios we also outperform ABE and Source-Val (see Table 5).

Table 4: Zero-Shot Generalization - Performance of NC (Top-1 Accuracy)

|  | **Scenario A:** max $Acc_{target}$ happens before max $Acc_{valid}$ | | **Scenario B:** max $Acc_{target}$ does *not* happen before max $Acc_{valid}$ | |
|---|---|---|---|---|
|  | ImageNet-S | ImageNet-R | ImageNet-A | ObjectNet |
| Oracle | 37.72 | 35.32 | 14.58 | 36.55 |
| Source-Val | 35.75 | 34.08 | 14.58 | 36.55 |
| Target-Val | 30.96 | 28.58 | 9.55 | 30.44 |
| PARC | 23.76 ± 3.12 | - | 9.74 ± 0.85 | - |
| NCE | 0.92 ± 0.00 | 16.67 ± 1.12 | 3.91 ± 1.12 | - |
| ABE | 24.66 ± 0.83 | 32.71 ± 0.30 | 8.34 ± 0.68 | 30.15 ± 0.97 |
| **NC** | **36.34** ± 0.04 | **34.32** ± 0.06 | 14.45 ± 0.01 | 36.51 ± 0.05 |

Table 5: Linear Probing - Performance of NC (Top-1 Accuracy)

|  | **Scenario A:** max $Acc_{target}$ happens before max $Acc_{valid}$ | | **Scenario B:** max $Acc_{target}$ does *not* happen before max $Acc_{valid}$ | |
|---|---|---|---|---|
|  | FOOD-101 | iNaturalist | PlantNet-300K | EuroSat |
| Oracle | 44.45 | 21.40 | 35.08 | 74.87 |
| Source-Val | 41.72 | 19.46 | 35.08 | 74.87 |
| PARC | 35.16 ± 0.36 | - | - | 34.43 ± 0.53 |
| ABE | 40.17 ± 0.62 | 17.23 ± 0.17 | 27.78 ± 0.13 | 63.47 ± 0.11 |
| **NC** | **43.80** ± 0.06 | **21.13** ± 0.04 | 35.08 ± 0.00 | 74.53 ± 0.02 |

A.1 ABLATION : WEIGHTING THE NEURAL COHERENCE ACROSS THE ENTIRE ARCHITECTURE

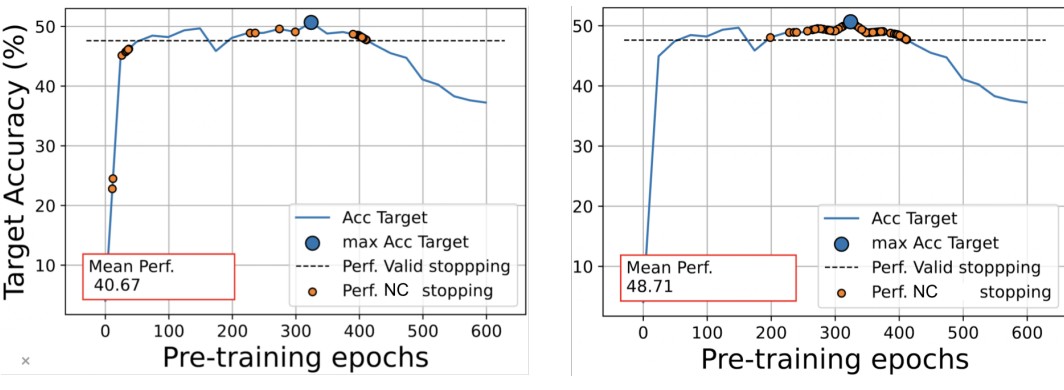

(a) Checkpoint Selection without weighting layers      (b) Checkpoint Selection with layer weighting

Figure 4: Multiple trials of checkpoint selection (orange dots) using a batch of 20 unlabeled examples. When the neural coherence of individual layers are not weighted (a) the high variance in estimated stopping time (orange dots) lead to a lower average target accuracy. When weighting the individual layers (b) the selected checkpoints are clustered around the global optimum (max Acc Target) and consistently outperform the checkpoint selected when relying on the source validation set (dotted line). This example shows ConvNeXt-L trained on ImageNet1k and Food101 [Bossard et al., 2014] as OOD target.

A.2 TRAINING DATA SELECTION

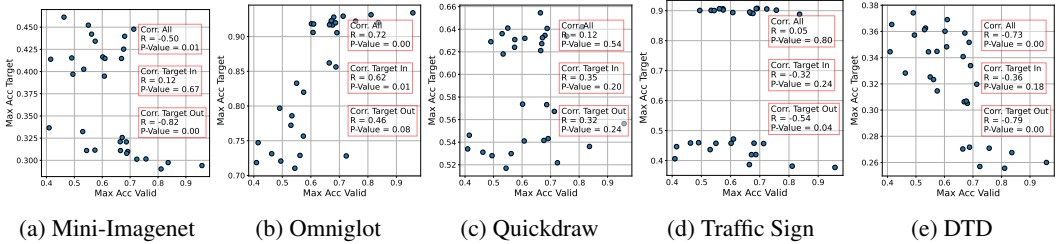

(a) Mini-Imagenet    (b) Omniglot    (c) Quickdraw    (d) Traffic Sign    (e) DTD

Figure 5: Source-Val Baseline : In-Distribution Validation performance does not correlate to target performance

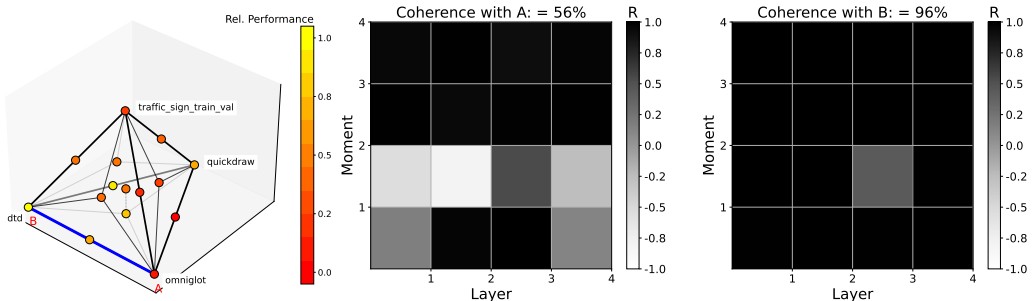

(a) Target Performance vs. Train (b) Neural Coherence indicates the best training data between A and B.
Data

Figure 6: Accuracies along the trajectory in the forward direction. Dataset A corresponds to Omniglot, Dataset B corresponds to DTD, and the Target Dataset is Mini-Imagenet. We will select the trained checkpoints of iteration 20500.

