# OpenReview forum: "Neural Coherence"
_ICLR.cc/2024/Conference — ICLR 2024 Conference Withdrawn Submission_

### Official Review · Reviewer_EvXa · 2023-10-25

**Soundness:** 1 poor
**Presentation:** 2 fair
**Contribution:** 2 fair
**Rating:** 3
**Confidence:** 4

**Summary:**

The paper studies the problem of model selection, i.e. finding the best pre-trained model for a given target task. To this end, the authors propose Neural Coherence (NC), an approach based on comparing trajectories of activation statistics between pre-training and target tasks. They show how this approach can be used to make model selection decisions for early stopping and pre-training/source dataset selection. Specifically, trajectories are constructed by ordering models based on the hyperparameter that should be chosen (such as the number of training epochs or the source dataset), computing approximations to the first and second moments of activation statistics across several samples and all layers of the models and comparing the angles between linear approximations to the resulting trajectory curves. The authors show empirically how NC can be used to select the best pre-training checkpoints as well as datasets.

**Strengths:**

- The problem of model selection for target tasks is quite relevant given the abundance of available pre-trained models and the lack of insights for choosing base models. The paper makes an interesting proposal for a model selection method that does not rely on annotations and seems to be very data efficient.
- The ideas behind NC seem reasonable, such as comparing statistics across different hyperparameter values and between the pre-training and target task.

**Weaknesses:**

- Soundness: The results don't seem convincing, given the complexity of the method.
    - The results seem unreliable: The proposed method makes a number of design choices and approximations, such as approximating trajectory comparisons using only the first two moments of activations, assigning higher weights to layers with more divergent statistics and summing up the stopping timesteps across layers. Given all of these choices, it seems easy for the method to overfit to the limited set of evaluation tasks. To make the conclusions more reliable, I would like to see results on:
        1. More target datasets, such as the ones in [A].
        2. More architectures. Right now there seems to be just one architecture used in 3.1 and one different architecture in 3.2. These two architectures are vastly different, the one in 3.1 uses > 200 hidden layers whereas the one in 3.2 has 4. The method seems to be sensitive to the number of layers, since activation statistics at different layers are considered with different weights, so studying understanding how the performance depends on this parameter seems quite important.
        3. Larger numbers of target samples. Most results are shown over 5 target samples with ablations using up to 20 target samples. While these numbers show that the method seems to perform well when target data is very scarce and unlabeled, a larger range of samples is needed to judge whether this performance advantage translates to perhaps more typical transfer settings with 100s and 1000s of target samples. In particular I would imagine the Target-Val baseline to perform much better in these cases. If this is not the case then the method should be advertised specifically for low-data regimes or in cases where target annotations are not available.
    - The results are not convincing:
        1. In Table 1 and 2, the proposed method (NC) performs 0.6% and 2% better than Source-Val, respectively, in Scenario A and slightly worse in Scenario B. Given that Source-Val is target dataset agnostic, i.e. it can just be computed once to find the best model on a source dataset and then use it on multiple target datasets, the added complexity of per-dataset evaluation of NC seems unjustified, given the at best marginal performance gains.
        2. Table 3 seems to report better results, but I'm not quite sure how to interpret them, see clarity issues below.
- Clarity
    - Motivation and justification for a number of methodological choices are missing, which makes the approach feel rather ad-hoc in many places.
        1. What is the intuition behind expecting the activation statistics (mean and std) to capture alignment between different tasks, especially when they are from different distributions? Why should models have similar activation statistics across these distributions?
        2. You mention that the moments of distributions are in many cases sufficient to characterize them. However, there are a couple of approximations, such as only using the first two moments, aggregating the first two moments of the mean activations and the first moment of the activation std, approximating trajectories linearly. How do these approximations affect how well NC captures the divergence between activation distributions?
        3. In Figure 1 you say that you sort the checkpoints in the trajectories based on performance (I'm assuming source data performance). However, in the "Training data selection" part of Section 2.2 you mention that you use the implicit ordering given by training time to sort checkpoints. The two criteria seem aligned but if there is a conflict, which one would you use?
        4. How did you choose the layer weighting scheme based on divergence described in 2.2?
    - Details for interpreting the results are missing:
        1. Which datasets or dataset combinations does NC select in Section 3.2?
        2. Why do you only train for 10% of the full pre-training steps in Section 3.2? Some of the datasets seem rather modest in size.
        3. In Section 3.1 you show two scenarios, one in which NC breaks down before the end of training (A) and one in which it doesn't until the end of training (B). How do you control for when divergence breaks down? Did you select the datasets based on their behavior in this regard?
        4. How is the Oracle constructed in Table 3?
    - Some notation is not properly defined
        1. What is $t^*_{valid}$ in Eq. 5?
        2. In 2.1 you mention an objective function $d_{NC}$, but do not define it, as far as I can see. Is $d_{NC}$ the same thing as $NC$ in Eq. 4?
    - It would be easier to understand NC if both it's ideal version, as well as the approximated form were presented, not just the approximation.

[A] Salman et al., Do Adversarially Robust ImageNet Models Transfer Better?, NeurIPS 2020

**Questions:**

- How does your method interact with Batch or Layer Normalization? These methods explicitly normalize the first and second moments across batch elements or features, so they try to remove (part of) the information that NC captures.
- How sensitive is the method to the choice of the source and target examples, esp. since you're using very small sample sizes? E.g. for pre-training checkpoint selection, what fraction of samples would select the best checkpoint?

### Minor
- Typos: there are a couple of typos, e.g. on page 5 "also working activation statistics" and a double "..", and on page 7 "of a full pretraining steps".
- Missing citation: In 3.2 you mention that you use the MAML algorithm for training models, it would be good to add a citation.
- In 2.1 you write "to reduce the signal-to-noise ratio", do you mean to increase the ratio?

---

### Official Review · Reviewer_zNiE · 2023-10-30

**Soundness:** 2 fair
**Presentation:** 1 poor
**Contribution:** 2 fair
**Rating:** 5
**Confidence:** 3

**Summary:**

The paper introduces a framework for predicting the performance of a pre-trained network on a downstream task. The proposed method, NC, relies on a small number of unlabelled samples from the target task, which are used to compute the gradient/slope (w.r.t. to a chosen hyperparameter) of a layer-wise statistic computed from the layer output feature vectors. This slope is then compared against that obtained with the source pre-training task to determine a divergence score. The effectiveness of the method is shown via evaluation on (1) selecting the best training checkpoint for zero-shot and linear probing performance on a different dataset, and (2) selecting datasets on which to pre-train for best performance on a downstream task.

**Strengths:**

- The proposed method performs well in low (target) data regimes, especially compared to directly evaluating on labeled target data.
- The method does not require any labels from the target task.
- It also appears to be computationally efficient, but it is not mentioned how many values of \Omega the summation term in equation (3) is computed over.
- The novelty of the method is difficult to ascertain (see weakness), but the method is applied to a new and important domain (model selection).

**Weaknesses:**

- Many portions of the paper are poorly presented, and implementation details are obscured (See questions) Also, title of the paper does not provide any insight about what the paper is about.
- The method proposed seems to be very closely related to [1], which also proposes the method of computing the same statistics $m_1$ to $m_4$ in the exact same manner, and modeling the trajectories of these variables. While compared against in the experiments, it is not mentioned in the related works nor anywhere else how the proposed method is different from that of [1]. Can the authors elaborate on the novelty/difference of their method compared to [1]?

- Experiments section:
   - Sec 3.1: The method seems to perform on-par with Source-Val especially for experiments on zero-shot generalization and linear probing (beats NC for 50% of the evaluated datasets), which does not require any data from or knowledge of any particular downstream dataset. This also renders Source-Val a more robust method since it generalizes to any unseen downstream task. In contrast, NC makes stronger assumptions regarding knowledge of (few samples) from the target downstream task.
   - Sec 3.2: The authors evaluate on 5-way-1-shot classification tasks. It is not mentioned what value of $n$ is used here, if $n=5$ as per zero-shot generalization experiments, this amounts to using the entire downstream dataset (minus the labels) which seems a strong assumption (compared to Source-Val with zero knowledge regarding the downstream task).
   - No baselines other than Source-Val are compared against in Sec 3.2 experiments, even though there exist several works for pre-trained model selection without labels such as [2]
- Other comments
   - An algorithm pseudocode for the generalization and model selection tasks might be useful for improving the clarity of the paper.
   - In the definition of $z = \phi_l(x; t)$, none of the notations $l$, $x$, and $t$ are introduced at this point (and $t$ is never introduced in the following text either)
   - Typo in: ``... by decreasing $L_S$, the target empirical risk $L_S$ will also decrease"



[1] Guiroy, Simon, et al. "Improving Meta-Learning Generalization with Activation-Based Early-Stopping." 2022

[2] Wallace, Bram, Ziyang Wu, and Bharath Hariharan. "Can we characterize tasks without labels or features?." 2021

**Questions:**

1. It is not clear what the top half of Figure 1 is showing, and it is not explained in the captions nor text
2. $z$ is treated as a random variable, but it is unclear what is the probability space on which this is defined. How is $p(z)$ computed?
3. Can the authors explain what they mean by: ``To avoid resulting higher order statistics, we only computes the 1st-order feature-wise moment over $m_2(z)$”?
4. What are the $\Omega$ values used for checkpoint selection ($\Omega_1$ and $\Omega_2$)?
5. [Sec 2.2: Training data selection] how exactly are these data distributions $p_A$, $p_B$ computed, and how is the interpolation between these two probability distributions done?
6. For which $l$ and which $k$ does the y-axis of Figure 2(b) refer to?
7. Sec 2, five concepts: ``Activation distributions embedded into a vector space” - it is not clear why this is necessary or ideal
8. It is stated that $n=5$ is used for the zero-shot generalization experiments, but value of $n$ used is not mentioned for linear probing and pre-training data selection experiments.
9. It is also not clear how the $n$ data points are chosen – are they chosen directly from the evaluation set, or from some separate training set? Are they selected in a stratified manner (across classes) or uniformly at random?
10. In Sec 3.2, selecting combinations of datasets for pre-training is mentioned several times, but it is not clear how these combinations are determined under the NC, or even Source-Val, framework.
11. Is Source-Val in Table 3 simply the best performing source dataset on its own task? So the same model is used to evaluate performance across all 5 target datasets (presumably Quickdraw?). How are compositional datasets evaluated?

---

### Official Review · Reviewer_KqcV · 2023-10-31

**Soundness:** 3 good
**Presentation:** 3 good
**Contribution:** 2 fair
**Rating:** 3
**Confidence:** 3

**Summary:**

This paper proposes a method called Neural Coherence that measures the activations of inputs throughout a network to characterize the behavior of a network on the corresponding inputs.  Empirically, the authors demonstrate that Neural Coherence can select good checkpoints to use for transferring a pre-trained model, and selecting the best pre-training dataset. Experiments are conducted in the vision domain on ImageNet-1K dataset evaluating on domain adaptation datasets like ImageNet-S and A, and transfer learning datasets such as Food-101 and EuroSat are used additionally.

**Strengths:**

* This paper proposes a new method for determining dataset and checkpoint transfer.  The proposed approach is simple to implement requiring only to save the activations for a subset of the data and training a linear regression on the activation trajectory making it broadly applicable to many applications and models, and requiring little overhead to implement. The authors demonstrate efficacy of the approach on large vision models (200M parameter ConvNeXt models).  The proposed approach outperforms prior work including PARC, NCE, ABE, and few-shot example selection.

* The proposed approach needs few data points to perform accurate checkpoint selection over prior works indicating its usability for constrained data settings where no target data or even source data is available, and can be used to shorten training for transfer reducing training computation costs.

* The proposed approach performs well on dataset selection. Achieving 4-10% performance improvement over the source-val baseline.  See Q1 below for clarification.

**Weaknesses:**

* The main weakness of the paper is that the experimental results are weak.  In particular results on zero-shot tasks (Table 1) do not perform much better than the Source-Val baseline.  This is consistent with findings in prior works that better ImageNet models transfer better: https://arxiv.org/abs/1805.08974 (i.e. doing well on in-dist validation set is a good measure for selecting checkpoints).  In Table 2, this is again apparent on EuroSat, however Neural Coherence is better on Scenario A by 2%.  In Figure 3, even though NC performs well with limited data, all curves are still below or similar to source-val.  The main advantage of NC for checkpoint selection seems to only be where you have limited target and source data as in this setting, source-val would also drop in performance.

* In checkpoint selection experiments, the authors the authors test different forms of fine-tuning including zero-shot and linear probe.  However, the authors do not test full fine-tuning.  The authors should conduct this investigation to show NC is consistently better for all benchmark choices of fine-tuning.

* The proposed approach is similar to https://arxiv.org/abs/2104.11408 which has applied a similar measure which they call Neural Mean Discrepancy (NMD) to compare activations and detect OOD data.  I encourage the authors to draw comparison with how the proposed method is different as the field of OOD detection is very similar to dataset selection.

Collectively, the method does not appear original in comparison with NMD, experimental results appear weak on checkpoint selection, and there are some outstanding concerns on dataset selection experiments concern small dataset size, and number of training steps.

**Questions:**

Q1: Why is the data selection experiment conducted only on small datasets for a subset of the number of training steps (10%)? A more standard setting might be to evaluate several pre-trained datasets/pre-trained models to select the best to transfer from based on NC. Further, why are there only 4 datasets for evaluation (25% chance accuracy).  Should this be 20% - or is the true training dataset not used for selection?

Q2: What do layer weights look like for datasets? Are there any consistent patterns in which weights contribute most to NC?

---

### Official Review · Reviewer_q2Kw · 2023-10-31

**Soundness:** 2 fair
**Presentation:** 3 good
**Contribution:** 2 fair
**Rating:** 3
**Confidence:** 4

**Summary:**

This paper involves tracking the "trajectory" of activations of the pre-trained model on the source and target data in order to pick the best pre-trained model for a task. Specifically, they claim that models with similar trajectories should be better for pre-training. They apply this technique to perform checkpoint selection and pre-training dataset selection.

**Strengths:**

I thought that the problem and the method were clearly stated. The method is original (I haven't heard of tracking the moments of the activations). The authors make a good point that the source-val (i.e., the best performing pre-training checkpoint) might not be the best checkpoint to perform transfer learning from.

**Weaknesses:**

**Problem Motivation**: I actually think the oracle baseline here (e.g., either performing zero-shot or fine-tuning the model on each of the candidate checkpoints on the target dataset) is a pretty reasonable way to solve the task. Especially since you are looking at linear probing, actually computing the linear probe for each of the checkpoints is already very cheap, so its not clear to me why you would need this method. Full-network fine-tuning is maybe a better case (where fine-tuning is expensive) but the authors do not consider this task in their paper.  Furthermore, in most cases, if you are about to perform transfer learning, you usually have a target dataset in mind (and have some non-trivial number of examples from that target dataset).

**Experiments**: In general, full-network fine-tuning is more common than either MAML or linear probing and should be considered in the paper. In particular, why was MAML chosen for 3.2?

**Intuition**: It's not clear to me (and not explained in the paper) *why* we should expect the trajectories between source and target to match for good transfer learning models. The authors should explain their intuition more clearly in the paper.

**Questions:**

How are the fixed set of target examples chosen for NC (5 seems very small)? Is it random? How sensitive is the method to the choice of the target examples? Does the source example used for NC need to match the target examples in some way?

For Table 3, are the source-val baselines just per dataset? (this seems unfair, since NC has access to combinations of datasets). Moreover, for this experiment NC seems to be significantly underperforming the oracle baseline. Do you hav an oracle baseline using the 10% checkpoints (trying to figure out if the gap is due to the 10% or the method).

How fast is this method (computing equation 1 seems pretty expensive)? Can you give some scalability numbers.

Do you have some examples from Scenario A (e.g., where choosing the last epoch isn't the best idea) where the images are substantially different from ImageNet? I could imagine that this scenario (where there is significant difference between the source and target dataset) would be hardest for this method.